# Adherence to DASH Dietary Pattern and Its Association with Incident Hyperuricemia Risk: A Prospective Study in Chinese Community Residents

**DOI:** 10.3390/nu14224853

**Published:** 2022-11-16

**Authors:** Kangqi Yi, Shuheng Cui, Minhua Tang, Yiling Wu, Yu Xiang, Yuting Yu, Xin Tong, Yonggen Jiang, Qi Zhao, Genming Zhao

**Affiliations:** 1Key Laboratory of Public Health Safety of Ministry of Education, Department of Epidemiology, School of Public Health, Fudan University, Shanghai 200032, China; 2Songjiang District Center for Disease Prevention and Control, Shanghai 201600, China

**Keywords:** hyperuricemia, dietary pattern, DASH diet, cardiometabolic diseases, cohort study

## Abstract

Hyperuricemia represents a great burden on global public health, and it is important to provide effective guidance at the level of dietary patterns. We evaluated the association between the Dietary Approaches to Stop Hypertension (DASH) diet and the risk of hyperuricemia in a large-scale, community-based cohort in East China. In total, 45,853 participants that did not have either hyperuricemia nor gout were included and assigned a DASH dietary score based on their baseline dietary intake. They were then divided into five quintiles (Q1–Q5) according to their score, followed by cross-linkages with local health information systems and in-person surveys. Cox proportional hazards models were adopted to calculate hazard ratio (HR) and 95% confidence intervals (CIs). During a median follow-up of 4.54 years, 2079 newly diagnosed hyperuricemia cases were documented. Compared to the DASH Q1 group, the risk of incident hyperuricemia for the Q5 group was significantly reduced by 16% (HR: 0.84; 95% CIs: 0.72–0.97) in the adjusted model. The associations of DASH diet with hyperuricemia appeared stronger (P for interaction <0.001) among participants with 3–4 cardiometabolic diseases at baseline, compared with their counterparts. Our results suggest that the DASH diet could be taken into account in the recognition of risk population and the prevention of hyperuricemia.

## 1. Introduction

Hyperuricemia, caused by purine metabolism disorder, represents a great burden on global public health. A high prevalence of about 20% was maintained during the period between 2007 and 2016 in the United States [1]. A similar prevalence was reported in Qatar, Iran, and Australia, ranging from 16.6% to 25.0% [2,3,4]. In China, hyperuricemia prevalence has significantly increased to 9.8–13.3% in recent decades [5,6], and even reached 25.8–34.05% in economically developed cities [7,8,9]. Moreover, data on the incidence of hyperuricemia remain sparse.

Hyperuricemia involves multiple systems associated not only with cardiometabolic diseases, but also all-cause mortality [10,11,12,13]. Previous epidemiology studies investigated the association between individual foods or nutrients and hyperuricemia, such as the intake of alcohol, red or processed meat, and sugary beverages [14,15,16,17]. However, most of them focused on a few foods or nutrients and tended to ignore the potential interactions between foods, nutrients, additives, contaminants, or unknown compounds [18]. Instead, considering the dietary pattern should form an integral part of the prevention and management of hyperuricemia, which is more appropriate for distinguishing high-risk groups and more instructive for a daily diet.

Dietary approaches to stop hypertension (DASH), a well-established dietary pattern, was originally developed to lower blood pressure. More recently, it has been associated with a lower level of serum uric acid (SUA) [19,20,21], but most published studies focused on a specific demographic, such as men or working populations. Prospective studies based on community residents are sparse. To address these limitations, we aimed to examine the incidence of hyperuricemia and evaluate the association between the DASH diet and the risk of hyperuricemia in a large-scale, community-based cohort in East China.

## 2. Materials and Methods

### 2.1. Study Design and Participants

The study is an ongoing, prospective, community-based cohort study conducted in China. Participants were recruited from four districts of Shanghai (Songjiang, Jiading, Minhang, and Xuhui) between April 2016 and December 2019 using a multi-stage, stratified, clustered sampling method. Twelve communities within these four districts were firstly selected as study sites, and then approximately one-third of the committees/villages were randomly selected from each community. All residents in these committees who were 20–74 years old and lived there for at least five years were invited to be participants at baseline. A standardized questionnaire survey and physical measurements were undertaken by trained interviewers and licensed physicians, respectively, to collect information on dietary intake, socio-demographic characteristics, lifestyle factors (smoking status, alcohol drinking status, and physical activity), and anthropometric information. Each participant also received biochemical examinations based on their urine sample and fasting blood sample by Dian Diagnostics Co. Ltd. (Hangzhou, China).

The follow-up information was mainly obtained from a cross-linkage with local health information systems via their national identification card, supplemented by in-person surveys. The local health information systems included an electronic medical record system and a death registry system, where the diagnosis of diseases was coded by the International Classification of Diseases tenth revision (ICD-10).

Of the 69,116 participants at baseline, in a sequence, we excluded 1408 participants with incomplete data upon the questionnaire interview, physical measurement, or biochemical examination; 2436 participants who reported any cancer; 3581 participants who had implausible values for total energy intake (<800 or >4200 kcal/d for men and <600 or >3500 kcal/d for women) [22]; and 2872 participants who had implausible values for other key variables. Among the remaining 58,819 individuals, 8470 participants who already had either hyperuricemia or gout at recruitment and 4496 participants who failed to link with local health information systems were further excluded. Therefore, a total of 45,853 participants were included in the current analysis (Figure 1).

### 2.2. Assessment of Dietary Patterns

The dietary intake during the previous year was assessed using a semi-quantitative food frequency questionnaire (FFQ) at baseline. This FFQ was specially designed and validated by thoroughly considering the traditional dietary habits in Shanghai, consisting of 29 groups of foods and beverages and 4 condiments [23]. For each food and beverage group, the intake frequency and amount consumed at each time was recorded and then multiplied together to calculate the average daily intake. There were six frequency choices, ranging from “never” to “3 or more times per day”. The amount consumed at each time was estimated using a quantitative method. Participants could either directly state the weight of the food they eat each time, or choose the number of portions per time based on standard tableware (bowls, plates, spoons, or cups). For each condiment, households frequently used packaging specification, and the use-up time was first determined to calculate the total daily food consumption of a family, and then, this value was divided by the number of family members eating at home. Nutrients and energy from each food group were calculated based on the Chinese Food Composition Table [24].

We then constructed the DASH dietary score based on dietary intake from baseline FFQ. We applied the method designed by Fung et al., essentially focusing on eight key components of foods or nutrients [25]. In particular, the component of low-fat dairy products was replaced by whole dairy products because of the relatively low intake of low-fat dairy products in China. For each component, all participants were classified into quintiles according to their intake ranking and then assigned a component score. Briefly, fruits, vegetables, nuts and legumes, whole dairy products, and whole grains are recommended in the DASH diet; therefore, their component scores were assigned 1–5 points in accordance with the order from the lowest to highest quintile. On the contrary, component scores for sodium, sweetened beverages, and red and processed meats were assigned 1–5 points in accordance with the order from the highest to lowest quintile since they are not encouraged. We then summed the component scores to obtain an overall DASH dietary score for each participant ranging from 8 to 40, where a higher DASH dietary score means a higher adherence to the DASH diet. Finally, participants were classified into five DASH dietary groups according to the quintiles of DASH dietary score from low to high (Q1 to Q5 group).

### 2.3. Assessment of Hyperuricemia

Hyperuricemia was defined as increased SUA above 420 µmol/L (7 mg/dL) in both men and women [26] or diagnosed by licensed physicians as hyperuricemia (E79) and gout (M10) in local health information systems.

The first in-person survey was conducted from April 2019 to August 2020 in Songjiang district. A total of 15,000 participants were randomly selected, of which 12,305 (82.03%) participants voluntarily completed the short questionnaire surveys, physical measurements, and biochemical examinations.

### 2.4. Assessment of Covariates

Smoking status was determined based on the response to the question “Have you ever smoked at least one cigarette a day for more than six months?”. Alcohol drinking status was determined based on the response to the question “Have you ever consumed alcohol at least three times a week for more than six months?”. Physical activity (PA) was assessed using the short form instruments of the International Physical Activity Questionnaire (IPAQ) and divided into three categories: “low level”, “moderate level”, and “high level” [27].

The history of cardiometabolic diseases was obtained through either self-report or an overall consideration of the results of physical measurements and biochemical examinations at baseline. Prevalent hypertension was defined as measured systolic blood pressure ≥140 mmHg, measured diastolic blood pressure ≥90 mmHg, or previous physician-diagnosed hypertension. Prevalent diabetes was defined as measured fasting blood glucose ≥7.0 mmol/L, measured glycated hemoglobin ≥6.5%, or previous physician-diagnosed diabetes. Prevalent hyperlipidemia was defined as measured triglyceride ≥1.70 mmol/L, measured total cholesterol ≥5.20 mmol/L, measured low-density lipoprotein cholesterol ≥3.40 mmol/L, measured high-density lipoprotein cholesterol <1.00 mmol/L, or previous physician-diagnosed hyperlipidemia. For each participant, the body mass index (BMI, kg/m^2^) was calculated based on measured height and weight and divided into three categories based on the Chinese reference standard for BMI: normal (<23.9), overweight (24.0–27.9), and obese (≥28.0).

### 2.5. Statistical Analyses

We calculated the person-years of follow-up from recruitment until the date when hyperuricemia or gout was first recognized in local health information systems, the date of in-person surveys if SUA met the required definition, or censoring at the date of death from other causes or the end of follow-up (31 December 2021), whichever was earliest. The incidence density and cumulative incidence rate of hyperuricemia were calculated.

The descriptive statistics of continuous variables were presented as mean and standard error, while categorical variables were presented as frequency and the column percentage. The difference in baseline characteristics between different DASH dietary groups was evaluated using the one-way analysis of variance or the $\chi^{2}$ test, as appropriate.

The multivariable Cox proportional hazards models were adopted to calculate the associations between DASH dietary groups and the risk of incident hyperuricemia. Initially, an unadjusted model was fitted. Then, a model minimally adjusted for baseline age (continuous) and sex (men or women) at baseline was fitted. Next, a fully adjusted model based on previous evidence was fitted to additionally adjust for marital status (married or not), education level (middle school or below, high school, and college or above), family history of hyperuricemia or gout (yes or no), BMI groups (normal, overweight, or obese), hypertension (yes or no), diabetes (yes or no), hyperlipemia (yes or no), smoking (never or ever), alcohol drinking (never or ever), PA (low level or moderate-to-high level), and total energy intake (continuous) at baseline. Finally, using the fully adjusted model, an analysis was performed by excluding participants who were diagnosed within 6 months and 12 months after baseline survey. The Schoenfeld individual test was used to test the proportional hazard. Conditional hazard ratio (HR) and 95% confidence intervals (CIs) of incident hyperuricemia for the Q2 group to Q5 group were calculated, compared with the Q1 group (the lowest adherence to DASH diet).

Subgroup analyses were conducted according to sex, baseline age, and cardiometabolic diseases. Sensitivity analyses were conducted using another definition of hyperuricemia to verify the robustness of results.

Our study conformed to Strengthening the Reporting of Observational Studies in Epidemiology (STROBE) guidelines. The level of statistical significance was set at two-tailed *p*-level of 0.05. All the analyses were carried out using R version 4.1.3 (R Development Core Team, Vienna, Austria).

## 3. Results

### 3.1. General Characteristics

Table 1 shows the baseline characteristics of participants according to the DASH dietary groups. Compared with participants in the lower DASH dietary group, participants in the higher DASH dietary group tended to be women, be younger, have a higher education level, and be engaged in more mentally rather than physically demanding work. However, participants in the higher DASH dietary group were less likely to smoke, drink alcohol, or have low-level PA; they also had a lower prevalence of overweight, obesity, hypertension, and diabetes.

We documented 2079 newly diagnosed hyperuricemia cases during a median follow-up of 4.54 years (196,550.40 person-years). The follow-up information of participants according to the DASH dietary groups is shown in Table 2. The incidence density was 10.58/1000 person-years (95% CIs: 10.13–11.03/1000 person-years) in total, while the cumulative incidence rate was 4.53% (95% CIs: 4.34–4.72%). The incidence density and cumulative incidence rate for men were 13.76/1000 person-years (95% CIs: 12.91–14.62) and 5.90% (95% CIs: 5.54–6.26%), respectively, and for women, they were 8.78/1000 person-years (95% CIs: 8.26–9.29) and 3.76% (95% CIs: 3.55–3.98%), respectively.

### 3.2. Association between DASH Diet and Hyperuricemia

Figure 2 demonstrates that the higher DASH dietary group was associated with a lower likelihood of the incidence of hyperuricemia. Compared with the Q1 group, the HR and 95% CIs for the Q2–Q5 group were 0.94 (0.82–1.08), 0.78 (0.68–0.90), 0.78 (0.69–0.89), and 0.62 (0.53–0.71) in the initially unadjusted model, respectively. A model minimally adjusted for baseline age and sex and another model further adjusted for other potential confounding factors were performed. These models showed that the risk of incident hyperuricemia was significantly reduced by 21% and 16% when comparing two extreme groups, respectively. Tests of trends were statistically significant in each model (*p* for trend < 0.001, *p* for trend = 0.004, and *p* for trend = 0.040, respectively). The results were similar when excluding participants who were diagnosed within 6 months and 12 months after the baseline survey (Table 3).

### 3.3. Subgroup Analyses

Figure 3 presents the results of subgroup analyses using a fully adjusted model. No significant interactions were found between the DASH diet and sex, age, BMI groups, hypertension, diabetes, and hyperlipidemia. However, a significant interaction was found between the kinds of cardiometabolic diseases at baseline survey and the DASH diet in relation to hyperuricemia. The association between the DASH diet and hyperuricemia appeared stronger among participants with 3–4 cardiometabolic diseases at baseline, compared with their counterparts.

### 3.4. Sensitivity Analyses

Sensitivity analyses were performed to verify the robustness of the results using another widely applied cut-off value of SUA (SUA increased above 420 µmol/L in men and above 360 µmol/L in women) [28]. The inversed association between the DASH diet and the risk of hyperuricemia did not materially change. The Q5 group was still lower than that in Q1 group with borderline statistical significance. Details are shown in Table 4.

## 4. Discussion

The current study documented an incidence density of 10.58/1000 person-years (95% CIs: 10.13–11.03/1000 person-years) using longitudinal data from 45,853 Chinese community residents. The inversed association between adherence to the DASH diet and the risk of hyperuricemia was found after adjustment for potential confounding factors. Previous incidence rates were reported to be 48.7–68.58 per 1000 person-years in South Korean and China [14,29,30], which were higher than current reports. The differences could be due to the following causes: Firstly, the majority of participants in the current study were women (63.91%). They are believed to have physiologically lower levels of SUA than men, potentially leading to a low overall incidence rate of hyperuricemia [31]. Secondly, it may be difficult to capture participants with asymptomatic hyperuricemia using local health information systems, since they rarely take the initiative to seek medical treatment (Appendix A). Comparatively, previous studies using physical examination data may distinguish them better. Finally, we used the new definition of hyperuricemia recommended by the Chinese Society of Endocrinology’s 2019 Guidelines. A higher cut-off level of SUA (7.0 mg/dL or 420 μmol/L), regardless of sex and age, also led to a relative lower incidence rate. Special attention should be paid to this discrepancy, which suggests that the disease burden of hyperuricemia problems may be just the tip of the iceberg, especially considering the differences in international guidance documents and the low diagnostic rate.

Few studies have examined the association between the DASH diet and risk of hyperuricemia based on a cohort of community residents. Our study supports the protective effect of the DASH diet on hyperuricemia, and using different definitions of hyperuricemia did not materially change the results. In previous studies, components which are not advocated for in the DASH diet, such as red and processed meats, were considered to be independently associated with the risk of hyperuricemia [17,32]. On the contrary, epidemiological evidence in dietary supplements suggests that glycine, tryptophan, and vitamin C contribute to the prevention of hyperuricemia. They are rich in vegetables, whole grains, nuts, and legumes, which are recommended in the DASH diet. Moreover, the anti-inflammatory effect of the DASH diet is beneficial for preventing hyperuricemia. A previous meta-analysis by Soltani et al. found that adherence to the DASH diet significantly decreases serum-sensitive C-reactive protein (CRP) levels in adults [25,33], while evidence from epidemiology studies shows that high-sensitivity CRP is positively associated with the risk of hyperuricemia [14,34,35]. Therefore, as a healthy dietary pattern, the DASH diet is a critical and intervenable factor related to the prevention of hyperuricemia.

Considering the significantly lower intake of dairy products in the Chinese population [36], this analysis of the DASH diet substituted low-fat dairy products with whole dairy products and revealed its protective effects (Appendix A). The inverse associations in dairy products with either the risk of gout or the serum urate levels were limited to the consumption of low-fat dairy products in some studies, which might arise from the counteracting effect of saturated fats [15,37,38]. However, some studies have indicated that the intake of whole dairy products could decrease the risk of hyperuricemia [16,39]. Antioxidant properties in milk are potential mediators of urate-lowering effects, such as lactalbumin, casein, and lipophilic vitamins [40]. Hence, an emphasis on the intake of dairy products at the level of dietary patterns does not necessarily change the beneficial effects of the DASH diet. In other words, the DASH diet focuses on the elements that it emphasizes rather than offering precise recommendations for the specific types of foods or the number of servings from each targeted food group [25]. The flexibility makes it possible to be adjusted according to the dietary habits of different countries.

An important finding in our study is the greater effects of the DASH diet on incident hyperuricemia in participants with 3–4 cardiometabolic diseases at baseline survey. Two possibilities may account for this disparate effect: On the one hand, participants with more cardiometabolic diseases tended to have a higher level of SUA (Appendix A), among whom the DASH diet showed a higher effect in the previous interventional studies [21,41]. On the other hand, the effect of improving insulin resistance and regulating renin-angiotensin in the DASH diet may play a cooperate role in reducing SUA concentration and thereby reducing the risk of hyperuricemia. The activation of the renin-angiotensin system and insulin resistance is associated with subclinical vascular injury and early kidney damage, which may increase the risk of hyperuricemia if the glomerular hyperfiltration cannot fully compensate for the reduced urate excretion [42,43,44]. Moreover, the DASH diet effectively resolves the contradictions in diet recommendations across different diseases. For example, traditional low-purine approaches overemphasize the limited intake of specific foods or nutrients, especially meat intake, which may lead to nutritional imbalance by replacing protein with unhealthy carbon-refined carbohydrates [45]. In this respect, what the DASH diet substantially emphasizes overlaps with existing studies of dietary factors, which are inversely associated the risk of common cardiometabolic diseases [46,47,48] and bring potential benefits. In summary, this finding highlights the need for promotion of the DASH diet, though further animal experiments and clinical studies are required to explain our results and the underlying mechanism.

To our best knowledge, this was the largest prospective-design and community-based study to date to examine this topic in China. Rigorous and adequate baseline data made it possible to adjust potential confounders and test the robustness of the conclusions. Moreover, cross-linkages with local health information systems minimized the loss to follow-up. However, some limitations need to be mentioned. Firstly, the self-reporting FFQ collected information on 29 foods groups instead of specific foods of each food group. The intake of foods and nutrients may be not completely accurate and recall bias may exist. However, in each food group, the intake proportions of component foods were calculated by its consumption divided by the total consumption of the food group using data from the Shanghai Food Consumption Survey (SHFCS) of 2013–2014. Hence, the FFQ fully considered the diet of local people and was shown to have good reliability and acceptable validity [23]. Secondly, it is possible that the incidence of hyperuricemia was underestimated, considering the fact that some asymptomatic hyperuricemia patients did not see a doctor. The combination with in-person follow-up may partly minimize this misclassification. Thirdly, diet and other covariates were based on data from the baseline survey, regardless of their changes during follow-up. As the in-person follow-up continues, more data could be applied to further studies to fully take these changes into account. Finally, the participants of this study were from Shanghai in eastern China, which may not be representative of the Chinese population as a whole. However, Shanghai is a mega-city associated with high urbanization, economic development, and early start of the Western diet [49]. It represents the changes in the dietary patterns of Chinese residents to some extent, indicating the practical significance of developing prevention guidelines for hyperuricemia on a national scale.

## 5. Conclusions

In conclusion, this prospective study highlights the incidence rate of hyperuricemia in the Chinese population and illustrates the inverse association between the DASH diet and the risk of hyperuricemia. This association tended to be significantly stronger among those with multiple cardiometabolic diseases. Our results suggest that the DASH diet could be taken into account in the recognition of the risk population and the prevention of hyperuricemia.

## Figures and Tables

**Figure 1 nutrients-14-04853-f001:**
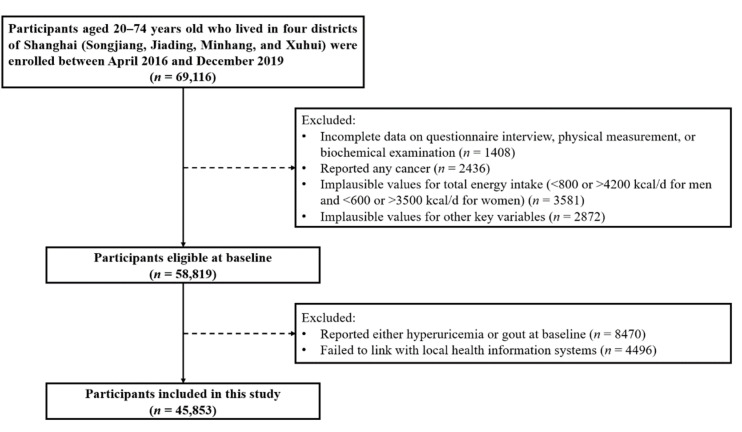
Flowchart of the study participants’ recruitment.

**Figure 2 nutrients-14-04853-f002:**
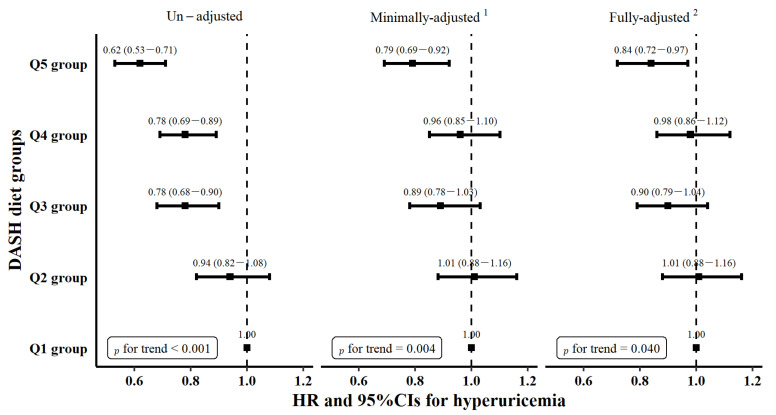
The association of DASH dietary groups with incident hyperuricemia. HR and 95% CIs were calculated with the use of the Cox proportional hazards regression model. ^1^ Minimally adjusted for baseline age and sex. ^2^ Fully adjusted for age, sex, marital status, education level, family history, BMI groups, hypertension, diabetes, hyperlipemia, smoking, alcohol drinking, PA, and total energy intake at baseline.

**Figure 3 nutrients-14-04853-f003:**
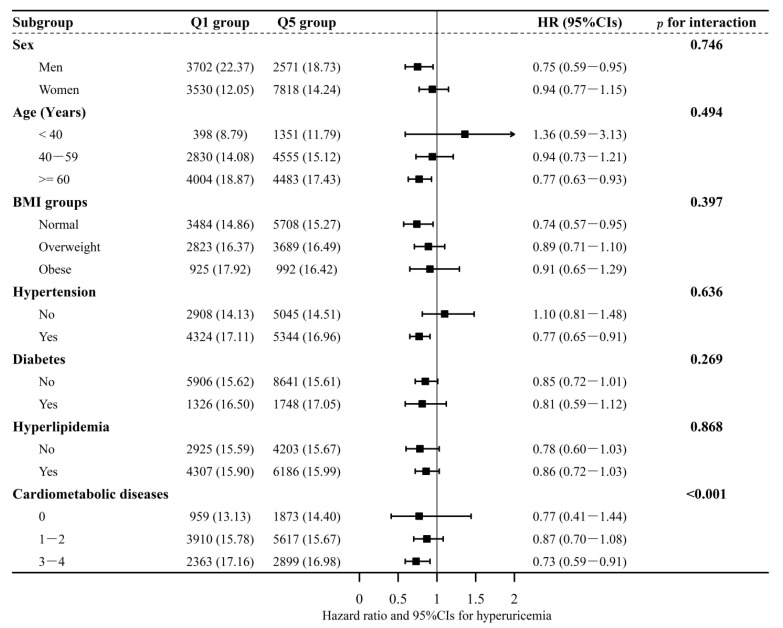
The association between DASH dietary groups with incident hyperuricemia in subgroups: a comparison of two extreme groups. The model was fully adjusted for age, sex, marital status, education level, family history, BMI groups, hypertension, diabetes, hyperlipemia, smoking, alcohol drinking, PA, and total energy intake at baseline. Each subgroup analysis was adjusted for all the covariates listed above except itself. Text in bold was used to indicate the summary value of subgroups. Arrows were used where the upper confidence intervals exceeds 2.00.

**Table 1 nutrients-14-04853-t001:** Baseline characteristics of participants according to DASH dietary groups.

Variables	Q1 Group	Q2 Group	Q3 Group	Q4 Group	Q5 Group	*p* Value
Socio-demographic factors						
Age (years)	58.98 ± 9.89	57.78 ± 10.48	56.48 ± 10.94	55.25 ± 11.54	55.34 ± 11.85	<0.001 ^1^
Men (%)	3702 (51.19)	3099 (42.62)	3280 (37.77)	3895 (31.73)	2571 (24.75)	<0.001 ^1^
Married (%)	6622 (91.57)	6727 (92.51)	8089 (93.14)	11,440 (93.2)	9655 (92.93)	<0.001 ^1^
Middle school and higher (%)	963 (13.32)	1200 (16.50)	1780 (20.50)	3318 (27.03)	3918 (37.71)	<0.001 ^1^
Family history (%)	350 (4.84)	314 (4.32)	377 (4.34)	652 (5.31)	546 (5.26)	<0.001 ^1^
Total energy intake (kcal/d)	2069.12 ± 714.59	2032.51 ± 685.11	2058.23 ± 672.55	2082.50 ± 648.65	2131.96 ± 597.05	<0.001 ^1^
Lifestyle factors						
Smoking (%)	2200 (30.42)	1703 (23.42)	1562 (17.99)	1603 (13.06)	916 (8.82)	<0.001 ^1^
Alcohol drinking (%)	1249 (17.27)	949 (13.05)	871 (10.03)	857 (6.98)	557 (5.36)	<0.001 ^1^
Low-level PA (%)	1319 (18.24)	1172 (16.12)	1292 (14.88)	1515 (12.34)	1002 (9.64)	<0.001 ^1^
Prevalence of cardiometabolic diseases						
Overweight (%)	2823 (39.03)	2845 (39.12)	3344 (38.5)	4549 (37.06)	3689 (35.51)	<0.001 ^1^
Obesity (%)	925 (12.79)	848 (11.66)	1027 (11.82)	1371 (11.17)	992 (9.55)	<0.001 ^1^
Hypertension (%)	4324 (59.79)	4286 (58.94)	4817 (55.46)	6502 (52.97)	5344 (51.44)	<0.001 ^1^
Diabetes (%)	1326 (18.34)	1370 (18.84)	1512 (17.41)	2081 (16.95)	1748 (16.83)	0.001 ^1^
Hyperlipemia (%)	4307 (59.55)	4331 (59.56)	5123 (58.99)	7142 (58.18)	6186 (59.54)	0.171

Data are presented as *n* (%), or mean ± SD. ^1^ *p* < 0.05. Abbreviations: kcal, kilocalorie; d, day; PA, physical activity.

**Table 2 nutrients-14-04853-t002:** Follow-up information of participants according to DASH dietary groups.

Variables	Q1 Group	Q2 Group	Q3 Group	Q4 Group	Q5 Group
Total No.	7232	7272	8685	12,275	10,389
Score ranges	11–19	20–21	22–23	24–26	27–36
Cases/person-years	430/32,565.41	402/32,365.33	389/37,995.57	532/52,318.77	326/41,305.29

**Table 3 nutrients-14-04853-t003:** The association between DASH dietary groups with incident hyperuricemia 6 months and 12 months after baseline survey.

Models	Q1 Group	Q2 Group	Q3 Group	Q4 Group	Q5 Group
Model 1					
Cases/person-years	415/32,560.42	391/32,360.82	377/37,991.68	507/52,313.05	310/41,299.58
HR (95% CIs)	1.00	1.02 (0.89–1.17)	0.91 (0.79–1.05)	0.97 (0.85–1.11)	0.83 (0.72–0.97) ^1^
Model 2					
Cases/person-years	398/32,537.76	371/32,339.10	358/37,974.35	480/52,285.36	292/41,284.18
HR (95% CIs)	1.00	1.01 (0.88–1.17)	0.90 (0.78–1.04)	0.97 (0.84–1.11)	0.83 (0.71–0.97) ^1^

Model 1 and Model 2 excluded participants who were diagnosed within 6 months and 12 months after baseline survey, respectively, both fully adjusted for age, sex, marital status, education level, family history, BMI groups, hypertension, diabetes, hyperlipemia, smoking, alcohol drinking, PA, and total energy intake at baseline. ^1^ *p* < 0.05.

**Table 4 nutrients-14-04853-t004:** The association between DASH dietary groups and incident hyperuricemia using an alternative definition.

Models	Q1 Group	Q2 Group	Q3 Group	Q4 Group	Q5 Group
Model 3	1.00	1.03 (0.90–1.18)	0.97 (0.85–1.11)	1.00 (0.88–1.14)	0.89 (0.77–1.03)
Model 4	1.00	1.04 (0.90–1.19)	0.98 (0.86–1.13)	1.01 (0.88–1.15)	0.89 (0.76–1.03)
Model 5	1.00	1.03 (0.89–1.18)	0.98 (0.85–1.13)	1.00 (0.87–1.14)	0.89 (0.76–1.03)

Hyperuricemia using an alternative definition: SUA increased above 420 µmol/L in men and above 360 µmol/L in women. Model 3 included all eligible participants, while model 4 and model 5 excluded participants who were diagnosed within 6 months and 12 months after baseline survey, respectively. All three models were fully adjusted for age, sex, marital status, education level, family history, BMI groups, hypertension, diabetes, hyperlipemia, smoking, alcohol drinking, PA, and total energy intake at baseline.

## Data Availability

Data described in the manuscript and analytic code will be made available from the corresponding author upon reasonable request.

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
