# Peer review of "Adherence to DASH Dietary Pattern and Its Association with Incident Hyperuricemia Risk: A Prospective Study in Chinese Community Residents"

_nutrients, 2022, doi:10.3390/nu14224853_

Round 1

Reviewer 1 Report

I found his study as a large, prospective-design, and community-based study to date to examine this association in China which can be replicated in other parts of the globe too. I suggest the acceptance of the manuscript in its present form with minor grammatical corrections.

Author Response

Dear Reviewer,

We sincerely thank for taking time out of your busy schedule to review the manuscript and give valuable feedback. The manuscript has undergone English language editing by MDPI, and we thoroughly checked the text for correct use of grammar again. The revised manuscript was uploaded as an attachment. Thank you again and best wishes to you!

Reviewer 2 Report

Table 1. set ascending data

Table 2. set ascending data

Table 3. set ascending data

Table 4. set ascending data

 Conclusions to change

Abstract: to be changed

Author Response

Dear Reviewer,

We sincerely thank for taking time out of your busy schedule to review the manuscript and give valuable feedback. The response letter was uploaded as an attachment. Thank you again and best wishes to you!
